# A Data-Science Approach for Creation of a Comprehensive Model to Assess the Impact of Mobile Technologies on Humans

Magdalena Garvanova [1,*], Ivan Garvanov [1], Vladimir Jotsov [1,2], Abdul Razaque [2,*], Bandar Alotaibi [3], Munif Alotaibi [4] and Daniela Borissova [1,5]

1 Department of Information Systems and Technologies, University of Library Studies and Information Technologies, 1784 Sofia, Bulgaria
2 Department of Cybersecurity, International Information Technology University, Almaty 050000, Kazakhstan
3 Department of Information Technology, University of Tabuk, Tabuk 47731, Saudi Arabia
4 Department of Computer Science, Shaqra University, Shaqra 11961, Saudi Arabia
5 Department of Information Processes and Decision Support, Institute of Information and Communication Technologies, Bulgarian Academy of Sciences, 1113 Sofia, Bulgaria
* Correspondence: m.garvanova@unibit.bg (M.G.); a.razaque@iitu.edu.kz (A.R.)

**Abstract:** Mobile technologies are an essential part of people's everyday lives since they are utilized for a variety of purposes, such as communication, entertainment, commerce, and education. However, when these gadgets are misused, the human body is exposed to continuous radiation from the electromagnetic field created by them. The communication services available are improving as mobile technologies advance; however, the problem is becoming more severe as the frequency range of mobile devices expands. To solve this complex case, it is necessary to propose a comprehensive approach that combines and processes data obtained from different types of research and sources of information, such as thermal imaging, electroencephalograms, computer models, and surveys. In the present article, a complex model for the processing and analysis of heterogeneous data is proposed based on mathematical and statistical methods in order to study the problem of electromagnetic radiation from mobile devices in-depth. Data science selection/preprocessing is one of the most important aspects of data and knowledge processing aiming at successful and effective analysis and data fusion from many sources. Special types of logic-based binding and pointing constraints are considered for data/knowledge selection applications. The proposed logic-based statistical modeling method provides both algorithmic as well as data-driven realizations that can be evolutionary. As a result, non-anticipated and collateral data/features can be processed if their role in the selected/constrained area is significant. In this research, the data-driven part does not use artificial neural networks; however, this combination was successfully applied in the past. It is an independent subsystem maintaining control of both the statistical and machine-learning parts. The proposed modeling applies to a wide range of reasoning/smart systems.

**Keywords:** signal processing; smart device; electromagnetic field; non-ionizing radiation protection; SAR; ANOVA; data science; selection; constraint satisfaction; preprocessing; mobile technology; machine learning; statistics

## 1. Introduction

The methods used in data science show ways to find solutions to a specific problem [1]. With the development of technology, the types of data to be analyzed are diverse and heterogeneous [2]. Having many and varied sensors to register an event or phenomenon is a great advantage in data processing and decision making but also a great challenge for data analysts [3]. The processing, aggregation, and analysis of disparate data is a complex process that can be facilitated by the use of intelligent solutions [4].

The study of the issue of the effects of smart devices on humans is an extremely recent scientific task that requires the processing and analysis of diverse data sets obtained from

numerous measurements [5]. It is not possible to give an unambiguous answer to this question when conducting the same type of research. Smart technologies are all around us and, in the near future, their number will increase many times over [6]. One of the most popular and currently used smart devices is the smartphone, which is used for both work and entertainment [7].

In recent years, this type of device has been increasingly used, and children and early teens own and use smartphones. These devices are used for communication, work, games, entertainment (watching movies and listening to music), visiting social networks, and more. At the same time, there are studies and analyses of the impact of these technologies on humans, and these effects are both psychological and biophysical in nature [8]. The smartphone is close to its owner, and the amount of time spent with this device is constantly increasing [9]. This process is difficult to interrupt or limit; however, if the consequences of overuse are studied and properly analyzed, the question of how to reduce the harmful effects on humans can be answered [10].

The psychological effects of smart technologies are the result of their long-term use and merging of the real and virtual worlds, which leads to social alienation, psychological loneliness, personal anxiety, low self-esteem, and hence to depressive states. More specific questions are related to: what is internet addiction and what physical and mental symptoms characterize this condition, the types of addictions, the extent to which they spread in society and what are the main areas affected, which is the most at-risk group among the population, what are the consequences, and—last but not least—the mechanisms of therapy and prevention.

Among the most commonly used methods of analysis are statistics from consulting agencies, content analysis of sites and blogs, and data from empirical studies and psycho diagnostic tests. The data analytics processes can be successfully combined with logic-based modeling instruments with the aim to create more medical applicable, versatile, and universal decisions.

Some experts find that dependence on smart technologies, and in particular on the services they offer, is not a separate behavioral disorder but a syndrome of a serious socio-psychological problem. The majority of researchers believe that the combination of addiction to cyberspace, together with electromagnetic radiation from smart devices, is a risk factor with dangerous consequences for the mental and physical health of an individual. Most smart devices communicate with each other using electromagnetic signals, which are a serious threat to human life and pollute the environment with invisible "electrosmog".

This article discusses the data-science approach to creating a comprehensive model for assessing the impact of mobile technologies on humans. The aim is to propose a data-science concept for the preprocessing, processing, and postprocessing of disparate data obtained from various sensors, measuring devices, and computer models for assessing the impact of mobile technologies on humans.

### 1.1. Paper Organization

The remainder of this paper is organized as follows. Section 2 presents the salient features of existing works. Section 3 presents the main statistical data processing methods. Section 4 shows the results from measuring the electromagnetic field, which reveal that, under certain conditions, mobile devices emit high-frequency electromagnetic waves and can cause various negative effects on humans. Section 5 discusses the most popular dosimetric values that estimate the levels of absorption of electromagnetic fields by the human body.

Modeling for SAR is used to mimic and illustrate the process of electromagnetic field absorption by the human head in Section 6. The collection and processing of thermal images are shown in Section 7. In Section 8, the experimental results and discussion are presented related to the change of brain activity of a mobile phone user. Section 9 proposes the use of complex data preprocessing, postprocessing, deep modeling, and analysis models by using intelligent methods. Finally, in Section 10, the paper is concluded.

*1.2. Research Methodology*

Extensive investigation, familiarity, and evaluation are crucial components in laying the groundwork for our suggested strategy. In order to handle and analyze heterogeneous data, we developed a sophisticated model based on mathematical and statistical techniques, and we then compared it to current state-of-the-art algorithms. In order to obtain these results, libraries were employed with existing algorithms. We reviewed the literature for a variety of study subjects and datasets and published the findings. These findings show that some of the outcomes are comparable to our suggested methodology.

The purpose of our study is to understand how mobile phones affect people in order to forward our efforts, which are described in this paper. In conclusion, because of the nature of the problem and the datasets that the algorithms are intended for, a true comparison is fairly challenging. In other terms, one may do better than the other in some situations, while the opposite results may occur in others. This article's focus does not enable for a thorough analysis and experimental investigation of each. A thorough evaluation of different methods is provided. Furthermore, it can be said that a more complex approach is required to solve the research issue, one that involves performing various experimental measurements, compiling statistical data, and using a computer model to explain some physiological effects brought on by electromagnetic wave exposure to the human head.

## 2. Related Work

This section discusses the main contributions of the current works. An assessment of the environmental and human health implications of base station and mobile phone radiation is provided [11]. A key invention that has changed people's lifestyles is the cell phone. With the widespread use of mobile phones in everyday life, the standard of living has significantly improved around the world. There have always been concerns about the effects of radio frequency radiation on humans, plants, and animals. Furthermore, it is alleged that the radiation emitted by mobile phones damages human health and jeopardizes the enjoyment and convenience derived from using the devices. The authors in [12] analyzed the changes that these smart phone technologies can bring to human–nature interactions while focusing on the outdoor behaviors of experienced outdoor users.

GHz [13] presented that the exposure of the human body to electromagnetic fields (EMF) with different frequencies can cause different biophysical effects. Thermal effects are typically minimal with frequencies less than 100 kHz; however, effects appear when increasing the frequencies. Smartphones communicate via high-frequency signals, and extended use of the generated electromagnetic field affects the skull. Additionally, irritability, memory impairment, weariness, anxiety, headaches, and disrupted sleep are primary indicators of changes in the body. It is believed that the changes caused by EMF are able to accumulate in the body under conditions of prolonged exposure.

As a result, pathologies, such as leukemia, brain tumors, and hormonal diseases, can develop. Research has investigated memory loss, Parkinson's and Alzheimer's disease, amyotrophic sclerosis, AIDS, and an increase in suicides in relation to EMF exposure [14]. Another consequence of exposure to EMF in people is the syndrome of premature aging of the body. Despite extensive investigations, there remain a variety of unknown and undiagnosed addictions in people induced by EMF. All of these impacts have been recorded using various research approaches.

These include the processing of thermal pictures to analyze thermal effects and the processing of EEG signals to assess brain activity [15]. To obtain a unified thorough evaluation of the impact of smart technology on humans, an intelligent approach for assessing various data is presented. To that end, this study proposes a framework for combining disparate data sets in order to assess the impact of smart technology on humans. Additionally, new methods for acquiring and evaluating empirical and experimental data are required to overcome the problem. A proposed paradigm for unification and intelligent solutions is effectively evaluated in this research for addressing difficulties.



### 3. Statistical Data Processing

It is feasible to collect data on the impacts of mobile devices on the psychological and physical health of the users to measure the impacts of active usage of smart technology. Correlation analysis was used to establish the relationships and the degree of dependence between individual variables. The most commonly used correlation coefficient is the Pearson coefficient ($r$) for linear correlation, which is calculated by the formula [16]:

$$r = \frac{P}{S_X S_Y} \tag{1}$$

where $P$—moment of the products; $S_X$—standard deviation of the variable $X$; and $S_Y$—standard deviation of the variable $Y$. The moment of the products ($P$) is calculated as:

$$P = \frac{\sum XY}{n-1} - \frac{\sum X \sum Y}{n(n-1)} \tag{2}$$

where $\sum X$—sum of $X$ values; $\sum Y$—sum of $Y$ values; $\sum XY$—sum of products of $X$ and $Y$; and $n$—sample size.

Another statistical criterion that is successfully used to determine changes in the responses and/or conditions of subjects as a result of an experimental intervention is Student's $t$-test for related samples, which involves research design "before-and-after". It has the ability to work with small volumes of data, and there is measurement "before the intervention", measurement "after the intervention", and the recording of statistically significant differences in the values of the tested variables. The empirical value of the t-test is calculated by the formula [16]:

$$t_E = \frac{|\bar{d}|}{\sqrt{\frac{\sum d^2 - n\bar{d}^2}{n^2 - n}}} \tag{3}$$

where $d = X_2 - X_1$ is the difference between two measured values of each object, $n$ is the number of observed objects, and $df$ are the degrees of freedom $df = n - 1$.

Among the most powerful statistical techniques for studying causal relationships is ANOVA (Analysis of Variance). One-way ANOVA provides analysis of the variation of a quantitatively dependent variable—for example, the degree of internet addiction caused by an independent qualitative or quantitative variable—for example, the age group. According to the null hypothesis, ANOVA is used to test the assumption of whether several means are equal, allowing determination of not only the differences between them but also which exact mean values differ from the others.

Table 1 shows the formulas for calculating the one-way ANOVA. The notations are as follows: $SSb$—sum of between-group squares; $SSw$—sum of within-group squares; $SST$—total sum of squares; $K$—degrees of freedom; $nj$—size (number of measurements) for each of the $k$ samples (groups); $\bar{x}_j$—sample mean for the $j$-th group; $\bar{x}$—total mean; $\bar{x}_{ij}$—mean of the $i$-th individual from the $j$-th group; and $n$—total sample size. From the presented Table 1, it is clear that the $F$-ratio is obtained by dividing the between-group mean square $MS_b$ by the within-group mean square $MS_w$:

$$F = \frac{MS_b}{MS_w} \tag{4}$$

Therefore, the logic of ANOVA is based on the decomposition of the total variance of the variable into two key components: the between-group variance (deviations of the group means from the total arithmetic mean) and within-group variance (individual deviations of the values from the mean within a category (group)).

**Table 1.** One-way ANOVA [16].

| Variance | Sum of Squares ($SS$) | $K$ | Mean Square ($MS$) | $F$-Ratio |
|---|---|---|---|---|
| Between groups | $SS_b = \sum_{j=I}^{k} n_j(\bar{x}j - \bar{x})^2$ | $k - 1$ | $MS_b = \frac{SS_b}{k-1}$ | $F = \frac{MS_b}{MS_w}$ |
| Within groups | $SS_w = \sum_{j=I}^{k} \sum_{i=I}^{n_j}(\bar{x}ij - \bar{x}j)^2$ | $n - k$ | $MS_w = \frac{SS_w}{n-k}$ | |
| Total | $SS_T = \sum_{j=I}^{k} \sum_{i=I}^{n_j}(x_{ij} - \bar{x})^2$ | $n - 1$ | | |

The Fisher's $F$-test is checked according to the significance level $\alpha$ (usually equal to or less than 0.05) and the degrees of freedom $K$, as follows: for the between-group variance $K_1 = k - 1$, where $k$ is the number of groups compared (degrees of freedom of the numerator); and for the within-group variance $K_2 = n - k$, where $n$ is the sample size (degrees of freedom of the denominator).

## 4. Measuring the Electromagnetic Field from a Smartphone

These measurements show the presence of electromagnetic fields generated by different models of GSM devices in different operating modes. Depending on where the measurements are made—outdoors or indoors, the results differ. The obtained values additionally vary depending on how far the smartphone is from the measuring equipment or from the distance to the base station, as well as what additional radio sources are nearby. For this purpose, experimental measurements were performed in which the GSM device was positioned at distances of 1 and 10 cm from the smartphone. The average measurement results are shown in Table 2. Measurements were obtained with Gigahertz HFE35C.

**Table 2.** EMF values generated by a GSM device.

| GSM Operating Mode | Distance between GSM and HFE35C [$\mu$ W/m$^2$] | |
|---|---|---|
| | 1 cm | 10 cm |
| **Outdoor** | | |
| Passive mode | 2 | 2 |
| Search mode | 520 | 150 |
| Conversation mode | 515 | 230 |
| **Indoor (Second floor of a building)** | | |
| Passive mode | 2 | 2 |
| Search mode | 640 | 220 |
| Conversation mode | 1550 | 530 |
| **Indoor (underground floor)** | | |
| Passive mode | 2 | 2 |
| Search mode | 860 | 310 |
| Conversation mode | Over 2000 | 1450 |

The data in Table 2 show that, in search and talk mode, the emission levels are many times higher than in passive mode. When ringing, the signal level is high for the first 20 s, and then decreases significantly. Depending on the location of the smartphone, the signal levels are different, and when indoors, the signal level may be above the normal values. The measurements were performed both indoors and in underground rooms, where the signal source from the base station was extremely weak, and, in order to achieve successful communication, the radiation level of the GSM device was at the maximum value to compensate for attenuation and to not let the conversation fail.

Studies have shown that the level of the electromagnetic field in urban conditions is many times higher than in rural areas. This confirms the assumption that the presence of different types of electrical devices and transmitters will lead to a significant increase in the background electromagnetic field. With the development of technology, this problem will



deepen and become more relevant. One of the most serious generators of electromagnetic fields is the smartphone. The level of the electromagnetic field generated by these strongly depends on the mode of operation and the environment in which the phone is located. In some cases, the levels of electromagnetic fields exceed the regulated permissible levels. The proximity of these devices to the human head requires in-depth study of the influence of electromagnetic fields on the possible effects on the human body.

## 5. Specific Absorption Rate

The specific absorption rate (*SAR*) shows how much radiation is absorbed by human tissues when irradiated by an electromagnetic field. The *SAR* is a measure of the rate at which the radio-frequency energy from a mobile phone is absorbed by the human body [17].

The local *SAR* is calculated as the power loss $dP_1$ absorbed in an infinitesimal mass $dm$ in the following way:

$$SAR = \frac{dP_1}{dm} = \frac{\sigma_{eff}E_{rms}^2}{\rho} = \frac{J_{rms}^2}{\rho\sigma_{eff}} \tag{5}$$

where $E_{rms}$ is the root mean square value of the electric field, $J_{rms}$ is the current density. $\sigma_{eff}$ is the effective conductivity of human brain tissue, and $\rho$ is the tissue density. Therefore, the *SAR* unit of measurement is W/kg. Energy from electromagnetic fields is absorbed into the human tissues and warms them. This leads to another definition of *SAR*, namely:

$$SAR = c_p \frac{\Delta T}{\Delta t} \tag{6}$$

where $c_p$ is the specific heat capacity of the tissue and $\Delta T$ is the change in temperature over a period of time $\Delta t$.

A distinction should be made between the instantaneous *SAR* and the permissible SARs, where an average value is measured for a given mass of tissue and a specified period of time. It is best to use a computer model to study the *SAR* and the thermal effects. The benefit of the model is that it visualizes the processes in depth.

## 6. Modeling for SAR Simulation

With the use of a computer model, it is feasible to thoroughly analyze *SAR* in the human body. This model may be used to research the effects of mobile phones on the human head since it can be used to visualize the processes of tissue absorption and heat deep within the human body. The characteristics of human tissues can be altered to imitate various age groups. The numerous tissues and bones that make up the human head model each have unique electromagnetic energy-absorption properties.

A mobile phone's characteristics can be changed to emulate various GSM device models. Numerous factors, including the antenna radiation pattern, transmitted signal strength, and signal frequency, are modifiable. The computer model's ability to see the depth of the human head, which can be utilized to analyze the absorption and warming processes in depth, is its most important feature.

For the proper functioning of the model, it is essential to know the biological characteristics of human tissues. The human body is composed of many organs and characterized by specific biological parameters that must be taken into account correctly in the model. The electromagnetic characteristics of the dielectric constant, magnetic permeability, and conductivity [18] should be properly defined for each modeled organ. The created computer model uses parameters that characterize the biological tissues of an adult. The shape of the human head was applied by the IEEE library and was loaded into the COMSOL Multiphysics software. The model was imported from a file named sar_in_human_head.mphbin [19,20].

The source of electromagnetic radiation was a model of a smartphone that was manually added. In the considered model, the location of the device was chosen to be on the left

side of the head in order to facilitate the comparison of the obtained results with thermal images from our previous experimental studies. The mobile device was modeled at a distance of 1 cm from the head as shown in Figure 1.

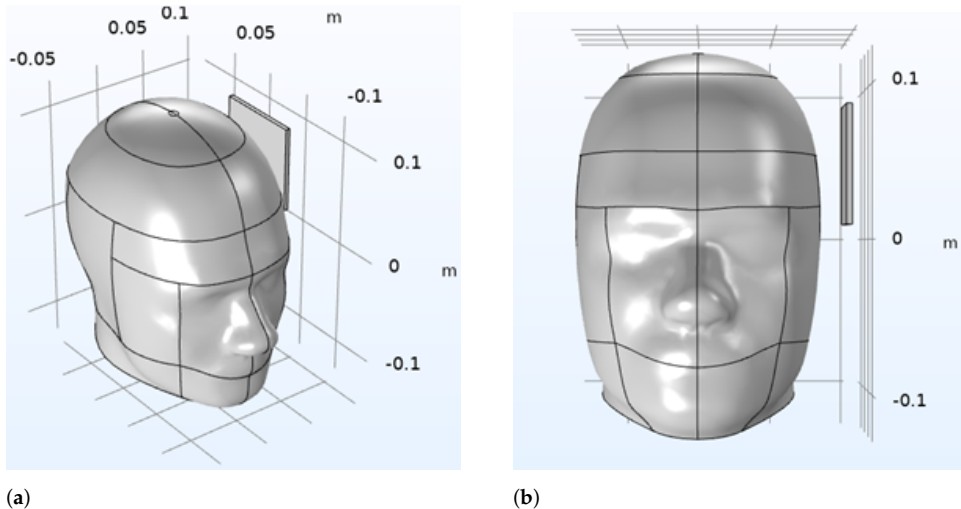

(**a**)                                    (**b**)

**Figure 1.** (**a**) and (**b**) Human head model and mobile device.

The electromagnetic parameters of the biological tissues of the human head were modeled through an interpolation function that uses the characteristics of the tissue inside the human head. The output for this function is directly taken from a file named sar_in_human_head_interp.txt. After simulation of the model, it is possible to estimate the *SAR* on any shape and tissue of the human body. When designing mobile devices, it is important to determine the amount of radiation that can be absorbed by the human body. The use of COMSOL Multiphysics and its radio frequency module allows a faster and more-efficient approach in the design of wireless devices that meet certain safety requirements. The local SAR value in the human head, calculated using the 900 MHz frequency equation, is shown in Figure 2.

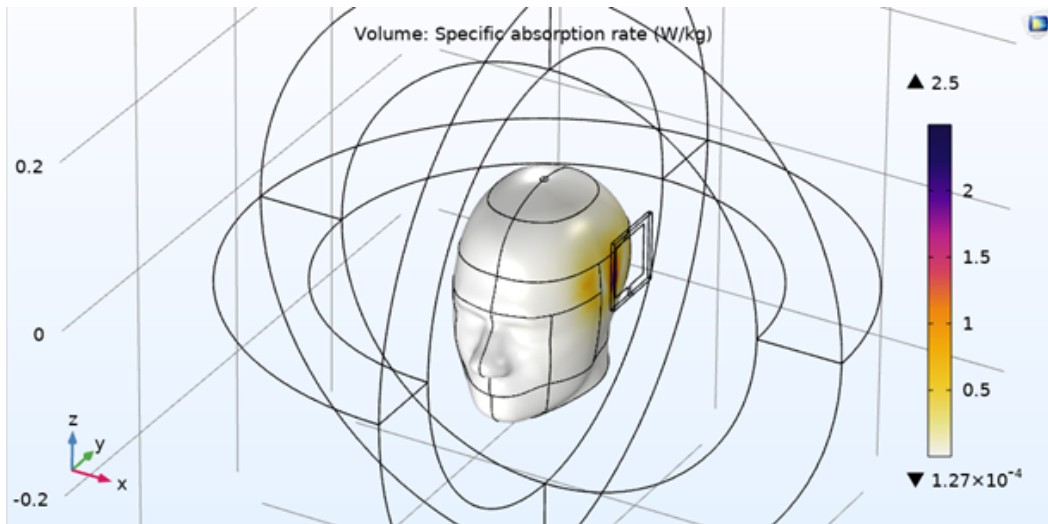

**Figure 2.** *SAR* visualization.

When talking to a mobile phone, the human head is very close to the phone, and the power of the emitted electromagnetic field is very high. Penetrating into a person's head, the electromagnetic field releases some of its energy, and the tissues in the head absorb this energy. Electromagnetic energy affects the particles in the tissue due to the

electrical and magnetic components of the electromagnetic field. Visualization of the effects of penetration of the electromagnetic wave into the human head can be shown by means of incisions of the head at certain levels (Figure 3).

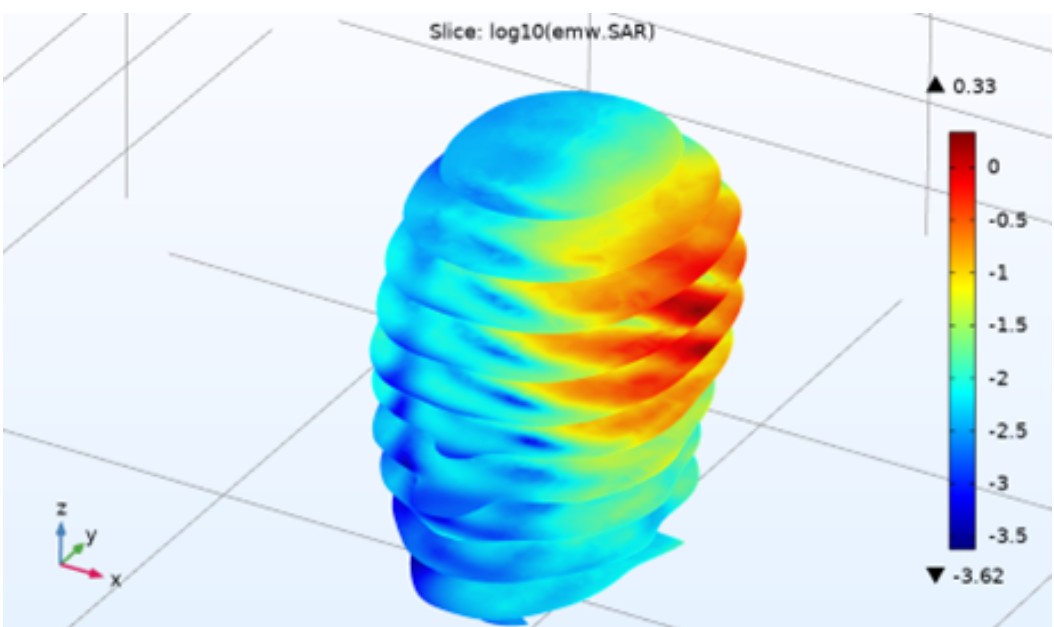

**Figure 3.** *SAR* visualization at different levels in the human head.

The strongest influence of the electromagnetic waves is in the head area, located in the immediate vicinity of the mobile device [17]. The greatest amount of energy is absorbed in this area, and the penetration into the human head is the greatest. The effects of exposure of the human body to radio frequency radiation mainly depend on the exposure time and the strength of electromagnetic fields. The penetration of the electromagnetic field into the depths of the human head depends on the frequency of the carrier signal. The higher this frequency, the faster it attenuates in space and the less it penetrates the human head. The highest values of absorption are observed on the surface of the human head. The developed model calculated only the local values of the *SAR* parameter. The maximum local *SAR* value is always higher than the maximum mean *SAR* value.

The amount of energy absorbed by the human head affects the temperature to which the tissues of the head are heated. The study of the processes of temperature distribution in the human head and on its surface is possible with the help of the created computer model using the COMSOL Multiphysics software. The frequency of the signal of the mobile device was selected to be 900 MHz. The transformation of the absorbed energy into heat was conducted with a biothermal equation. The change in temperature is a function of the physiological properties of biological tissues and blood circulation in the human body [21]. The thermal effects on and in the human head are shown in Figure 4.

Due to the created computer model, it is possible to study the processes of penetration of the electromagnetic field into the human head and the effects caused by this, thus, thoroughly simulating different situations and different characteristics of the head model and mobile phone characteristics. The visualization capabilities of the COMSOL Multiphysics software are impressive and allow a detailed view of the simulation results.

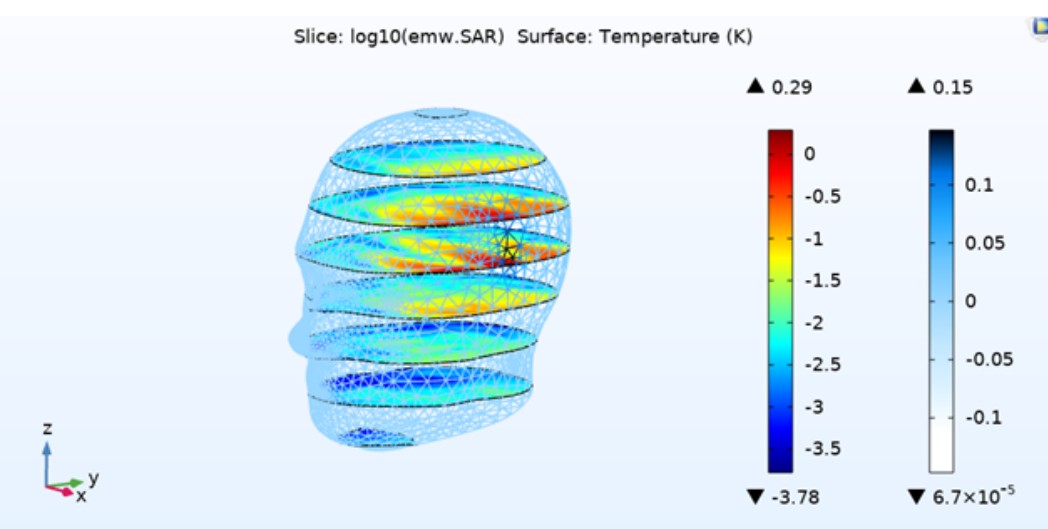

**Figure 4.** Visualization of the thermal effects in the human head for several horizontal layers.

## 7. Collection and Processing of Thermal Images

The impacts of the use of GSM devices on the physical condition of a person can be assessed by the thermal effects caused by the electromagnetic waves emitted by GSM devices. The experimental scenario includes a speaker with a GSM device for 20 min and a FLIR P640 thermal camera to capture their head in profile and full face. During the conversation, the GSM device was located about 1 cm from the head of the participant in the experiment, and the thermal camera was about 2 m away, focused on their head. The average room temperature was around 22 °C.

As a result of the 20 min conversation and the irradiation with the electromagnetic waves from the GSM device, the head of the participant in the experiment warms up by about 1–2 °C as can be seen from Figure 5. The increase in head temperature depends on the duration of the conversations. When talking for up to 30 s, no change in intracranial temperature is observed; however, when talking for more than 2 min, first, the ear begins to warm up and then the soft tissues around the ear. The increase in temperature is a result of prolonged irradiation of the human body with high-frequency radio signals from the GSM device.

The obtained results show that the temperature on the surface of the head is the highest and decreases in depth. The temperature change near the mobile phone is on the order of 0.6 °C and decreases rapidly inside the head. The thermal effects obtained from the model largely coincide with the results of a real experiment conducted with a thermal camera (Figure 5).

Averaging the temperature of the head before and after the experiment resulted in a temperature difference of about 1.3 °C. After checking the number of pixels exceeding the temperature of 34 °C before and after the experiment, we found that, after the experiment, the area heated above this value was three-times larger than before the experiment. The thermal images were processed using MATLAB. The study found that the temperature of the head on the side of the GSM device heated up much more than the other side. The areas around the ear, forehead, and neck heated up much more than the rest of the head.

The study of the processes of penetration of the electromagnetic field into the human head and the effects caused by this is an extremely important scientific task. The interaction between the human head and the electromagnetic radiation caused by cell phones can cause electric currents and electric fields in the human head, which can lead to negative health effects.

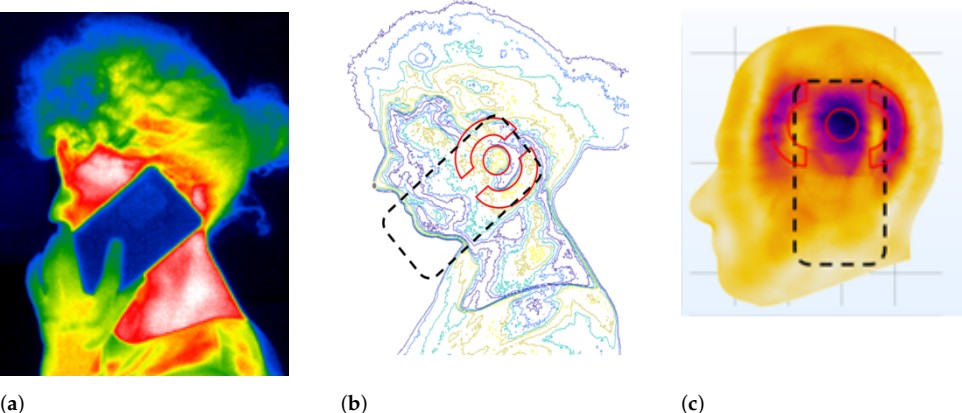

(**a**)             (**b**)             (**c**)

**Figure 5.** (**a**) Visualization of thermal effects in a human head before use of a GSM device; (**b**) visualization of thermal effects in a human head during use of a GSM device; and (**c**) visualization of thermal effects in a human head after use of a GSM device.

## 8. Experimental Results and Discussion

Electroencephalographs (EEGs), which measure electrical signals generated by the brain (brain waves), are used to study a person's brain activity. EEG signals are obtained as a result of the work of neurons in the human brain and can be intercepted using electrodes attached to the surface of the scalp [22–24].

A series of experiments were conducted to analyze the possible effects of the electromagnetic fields generated by smartphones on the activity of the human brain. Thirty volunteers (16 men (53.3%) and 14 women (46.7%) with an average age of 45.2 years) participated in the studies; however, we plan to increase these numbers in future studies among adults and children. The participants in the study stated that they were physically and mentally healthy, that they had not taken any medication before the tests, and that they were voluntarily undergoing these tests. The experiments were conducted in two stages.

The first stage involved studying the EEG signals of the subjects without using a mobile phone. The second stage of the experiment was performed while the subjects used a mobile phone (Figure 6). The EEG recordings from the two experiments were processed in the MATLAB environment in the time and frequency domain of the signal. The aim was to make a comparative analysis of the signal spectra from the two experiments.

Measures to reduce any other brain activity have been taken to assess the effect of cell phone electromagnetic radiation on a person's brain activity. The experiments were conducted in a quiet and cozy room, with the test subjects placed in comfortable armchairs with their eyes closed to reduce side stimuli. During the experiments, participants held the phone at a distance of about 1 cm away from the head, listening to a quiet countdown from one to one hundred, which was started by a researcher in another room. The aim of the experiment was to be as close as possible to a real conversation as shown in Figure 6.

A mobile phone with a SAR of 0.36 W/kg was used during the experiment. The experiment lasted about an hour with the first 30 min without a phone and the second 30 min with a phone. The obtained signals were filtered and divided into frequency ranges, respectively: delta $\delta$ (1–4 Hz), theta $\theta$ (4–8 Hz), alpha $\alpha$ (8–13 Hz), and beta $\beta$ (13–32 Hz). With the help of the Pwelch function of the MATLAB program, the spectra of the signals before and after a call with a mobile device were obtained. The spectra of the two experiments for all electrodes were compared, and differences in the spectra were found at several measurement points (Figure 7).

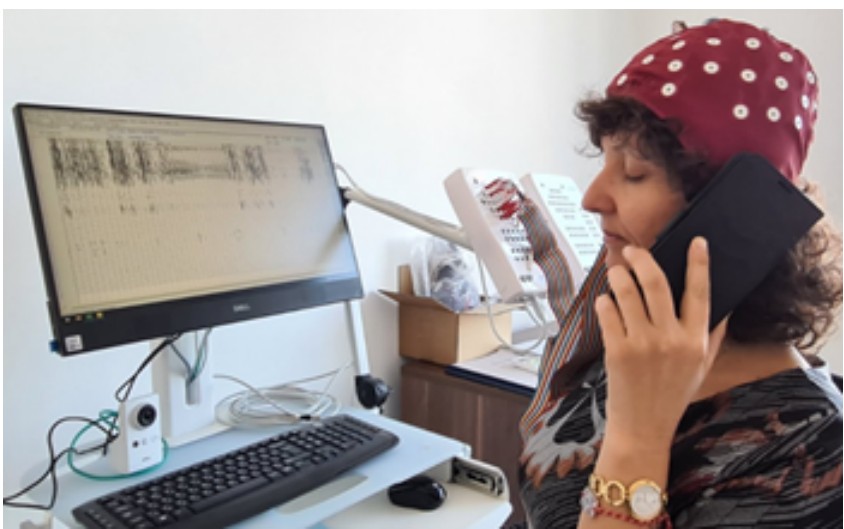

**Figure 6.** Participant during the experiment.

At the points with the numbers T3, T5, and F7, which are the closest to the mobile phone, a significant change in the spectral activity of the brain was found. The largest change in the spectrum was found in T3, where the changes were in the theta, alpha, and beta frequency ranges. The changes in the spectra at points T5 and F7 were only in the theta and alpha ranges. Interestingly, this dependence was found in all participants in the experiments but to varying degrees.

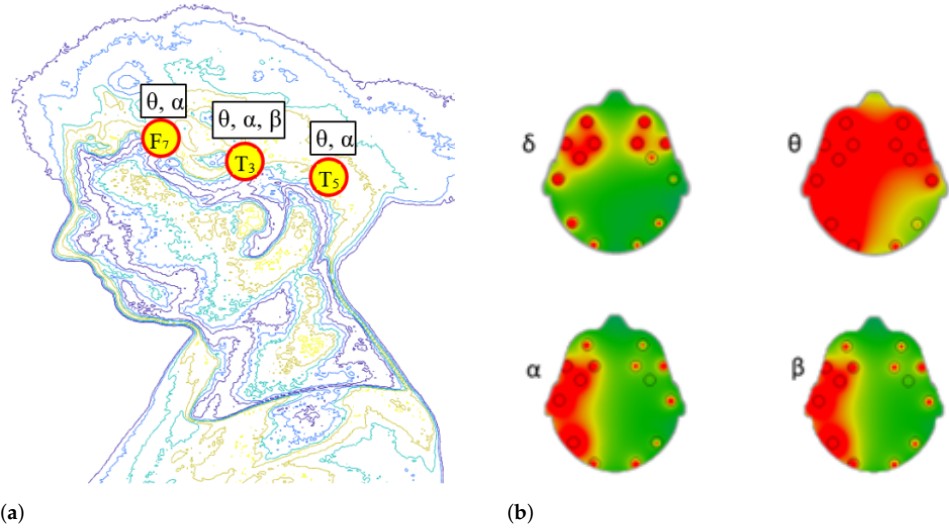

(**a**)　　　　　　　　　　　　　　　　　　　　　　　　　(**b**)

**Figure 7.** (**a**) Differences in the spectra by range and (**b**) different GSM ranges for delta, theta, alpha, and beta.

To compare the average spectral exposure with and without GSM for the ranges of delta $\delta$ (1–4 Hz), theta $\theta$ (4–8 Hz), alpha $\alpha$ (8–13 Hz), and beta $\beta$ (13–32 Hz), Student's *t*-test for related samples was used (Howard, 2008). Statistically significant differences, where $p < 0.05$, are visualized in Figure 7.

The change in brain activity in a person's head on the side of a mobile phone has a short-term effect that is shown to be due to the operation of a mobile phone. If the *SAR* is studied and analyzed in more detail using a computer model, we expect that the changes in brain activity will be closely related to the location and amount of absorbed electromagnetic energy. This relationship has not been studied in sufficient depth and requires further research into the body's biophysical responses; thus, this is of interest for future research.

*Accuracy*

The degree of similarity between a measurement and its real value is referred to as accuracy. A limited number of EEG channels recorded concurrently can improve the accuracy. Two distinct types of tests were conducted to evaluate brain activity and lasted 30 min. The first experiment was performed without a cell phone (GSM); however, the second experiment included a mobile phone while subtracting brain activity. Interesting discoveries were made, and it was revealed that, while utilizing a cell phone, the accuracy was somewhat reduced.

Figure 8 demonstrates the accuracy and compares the average spectrum exposure with and without GSM for the ranges of delta (1–4 Hz), theta (4–8 Hz), alpha (8–13 Hz), and beta β (13–32 Hz). Figure 8a indicates that the accuracy without a mobile phone was 99.95%, whereas the accuracy with a mobile phone was 99.78% utilizing a delta range of (1–4 Hz). Figure 8b shows that, with the theta (4–8 Hz) frequency range, 99.88% accuracy was obtained without a mobile phone, whereas 99.69% accuracy was obtained with a mobile phone.

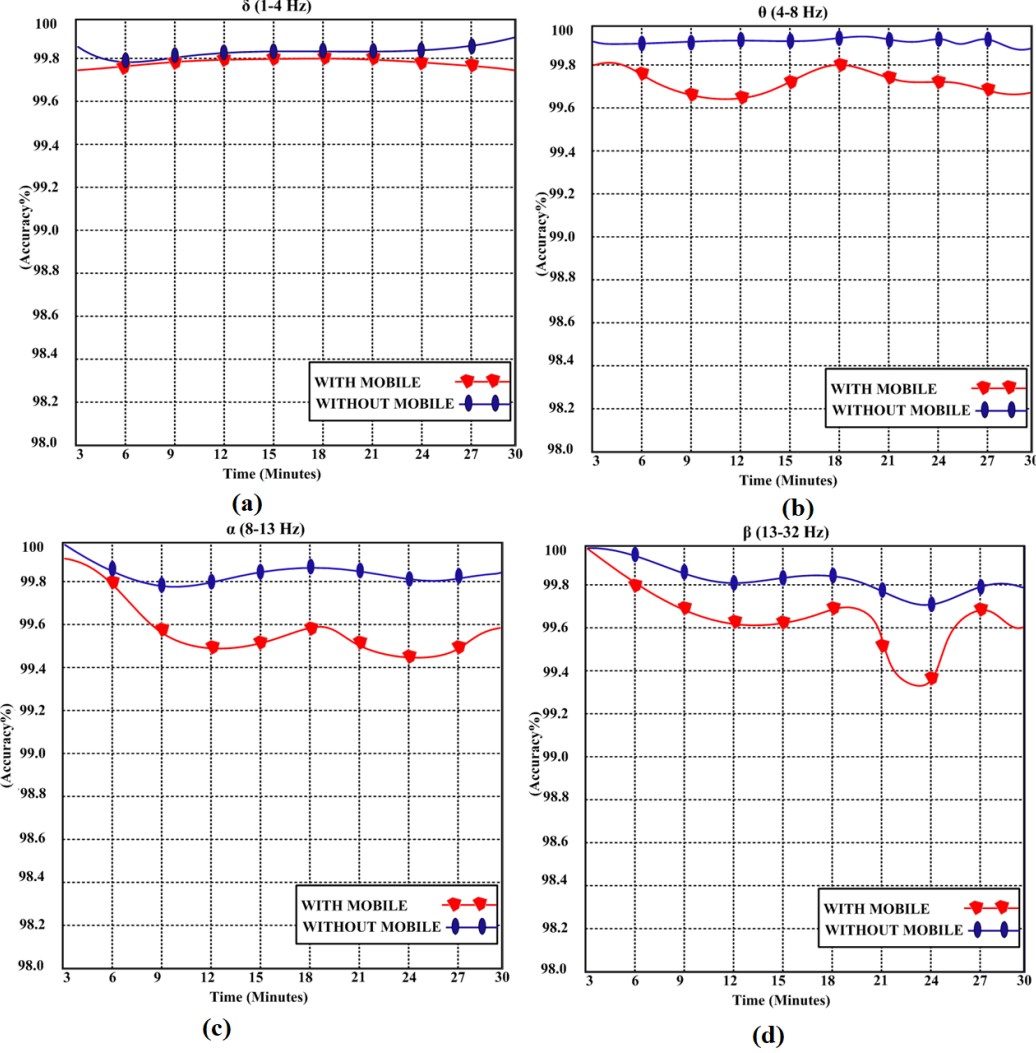

**Figure 8.** (**a**) The accuracy of the average spectrum exposure with and without GSM using the frequency range of delta δ (1–4 Hz). (**b**) The accuracy of the average spectrum exposure with and without GSM using the frequency range of theta ϑ (4–8 Hz). (**c**) The accuracy of the average spectrum exposure with and without GSM using the frequency range of alpha α (8–13 Hz). (**d**) The accuracy of the average spectrum exposure with and without GSM using the frequency range of beta β (13–32 Hz).

Figure 8c depicts the 99.81% accuracy with the alpha (8–13 Hz) range in the absence of a mobile phone. A cell phone, on the other hand, achieved 99.58% accuracy in the same frequency band. Figure 8d exhibits 99.80% accuracy without a mobile phone utilizing a beta (13–32 Hz) frequency range and 99.59% accuracy with a mobile phone. It was shown that the GSM had a negligible impact on the signal accuracy while monitoring brain activity.

## 9. Processing of Complex Data Analysis Models Using Intelligent Methods

Intelligent methods are the processes of gathering, modeling, and analyzing data in order to derive insights that may be used to make decisions.

### 9.1. Deep Knowledge Modeling and Constraint-Based Fast Preprocessing

The preprocessing of data is effectively utilized in different domains, such as number theory, cryptography, intelligent measurement, education, and bioinformatics. The chances of making an improvement will be quite slim without a comprehensive description of the surroundings. It is clear from working with description logic that logic-based modeling is challenging to control, that it is challenging to merge the logical and statistical phases of the data science cycle, and that their algorithmic complexity is often considerable.

On the other hand, it is possible to reason using the body of knowledge, and this capability enables the development of knowledge- and data-driven open systems. It is simple to combine the suggested study with the mentioned non-classical logical methodologies. More information is available in [25]. When discussing data-driven methodologies, artificial neural networks (ANNs), such as deep learning, are frequently used. In order to enhance the quality of human-like reasoning, this paper investigated a novel data-driven methodology that makes use of modeling and constraint fulfillment characteristics.

The constraint satisfaction methods are frequently used in data-preprocessing issues. A selection of data was applied aiming to complete preprocessing more efficiently, and the considered deep-modeling constraint satisfaction methods significantly improved this process. Currently, the following groups of novel logically-inspired constraints have been investigated in [25–27]. The mema method for control is named Puzzle but it significantly differs from the methods constructed for solving puzzles, such as [26–29].

The latter are ineffective because of the usage of random number strategies. The proposed Puzzle approach is easily combined with these and other methods [30] with the aim to increase their efficiency. Initially the classical constraint satisfaction methods can be applied with the purpose to form a closed focus area where certain data may be logically connected.

Let us focus on the considered two objects, ones of the many enclosed in the area from Figure 9. The closed 'focus/selection area' helps to reveal new knowledge concerning the enclosed objects. The data analysis concerning this case can reveal new knowledge, for example, that $M$ implies $N$ or that $M$ has some relationship to $N$. The second case is the result of the classical link analysis and/or corresponding data mining applications. Some general disadvantages: the classical approach works with a priori given data and is not intended for the elaboration of open systems.

In practice, the constraints should be dynamically changed depending on the current knowledge/data. This becomes possible after the introduction of new types of non-classical constraints. Generally speaking, there should be new types of constraints introduced. Constraint violation is impossible in the classical case but this case should be reconstructed. Every rule/constraint could be defeated depending on the conditions. The new types of constraints make it possible. Somebody who exposed their body to radiation generally does not think about cancer that could occur 20 years later. Many other application problems arise in medical practice where the problems are gradually accumulating.

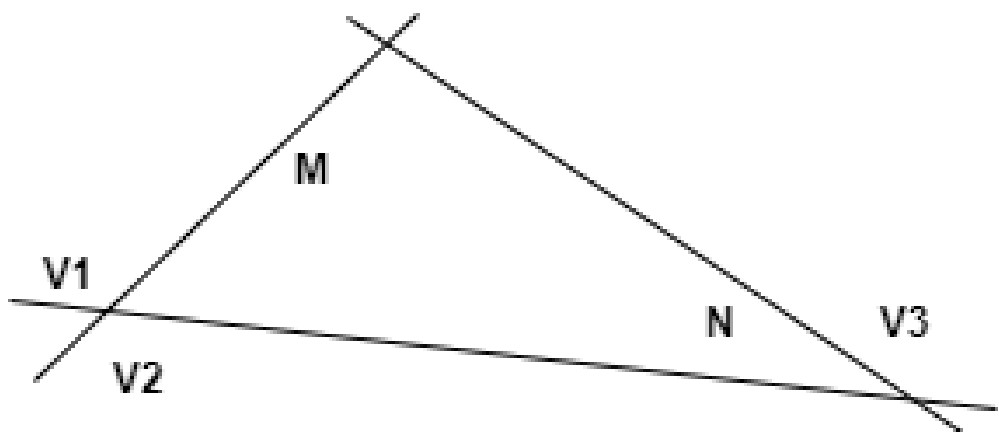

**Figure 9.** A set of three linear constraints constructs a closed area focusing on objects *M* and *N*.

With new conditions, additional questions arise: **why** the constraints are imposed, **what** and when violates it, **where** it could be defeated, and other use cases.

Binding, pointing, and crossword are the new groups of constraints. There exist many binding situations, and some of them have been researched in this article. The first case is when the maximum binding possibility is concentrated in the center of the area, and the distance from it is a function diminishing its value. The proposed research revealed that the binding may depend on certain conditions, and it may also influence the features of all objects contained in its area: the linearity/range/the region of the usage and/or other properties could be defeated. The general case is depicted in Figure 10 where curves *B* and *D* are the bindings concerning the searched goal *G1*.

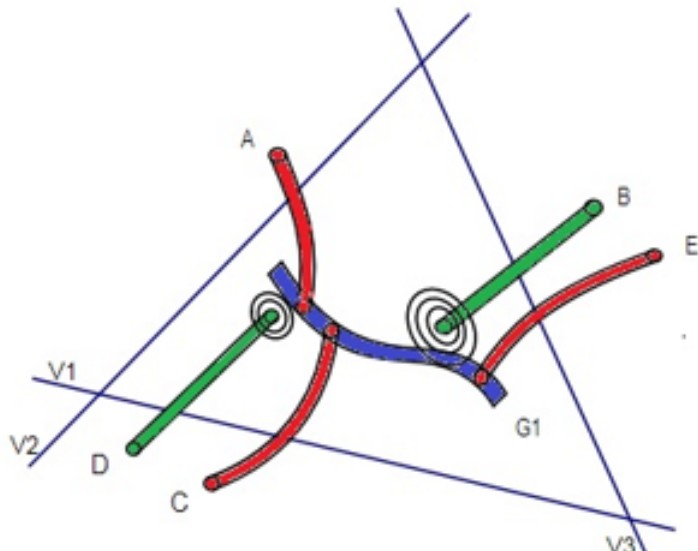

**Figure 10.** A set of nonlinear constraints in combination with the proposed three groups of logically-based constraints.

Association methods are generalized in binding constraint theories [31–36]. Associations are purposed for finding implicative classical form of rules, while the binding constraints explore any form of causal relationships. We investigated only a few binding constraint groups but they were very useful as a data-science modeling tool.

1. Denote *A* is bound to *B* if there is proven evidence of any form of causal relationship between them. The implication is also included in this case.
   Should the type of the causal form should be described in special forms of meta-knowledge attached to the corresponding binding case? For example, some people

are very sensitive to long phone calls. The personal binding constraint 'phone call > 10 min'− >'tired' or 'noise in the head' should be added to the modeling case. The metaknowledge should include the complex of disease history, a nervous state, a history of complaints, and corresponding factors. Semantic reasoners are very helpful in binding information processing.

2. A frequently used form of binding is 'the solution to the problem is somewhere here'. This case is depicted in *B* and *D* areas in Figure 9. In this case, type-1 fuzzy systems are effectively used in combination with binding constraint modeling. The most possible solution zone is situated in the binding very center. Accordingly, the distance from the center diminishes the possibility to bind the corresponding features. For example, the center of this binding area could be the area where is the pain located.

3. The above quoted cases did not influence the other modeling methods schematically depicted in Figures 9 and 10. In many cases, this is not enough. In this case, the application of the binding zone introduces defeasible reasoning in the zone and/or around it. The proposed forms of defeasible reasoning are discussed in [37], and the introduction to this research is briefly presented below. In our medical modeling practice, every part of knowledge could be defeated and/or exchanged with other knowledge forms.

4. The cases 1–3 are unconditional cases. In practice, many binding relations exist only in some special conditions: if $|T_1 \star T_2 \cdots T_k$ are true, then *A* is bound with type 1, 2 or 3 to *B*, where $\star$ denotes a logical operation conjunction or disjunction. The corresponding $T_i$ are a priori given.

5. The variant of type-4 binding was considered where the binding control is data-driven and $C_i$ are formed by using ANNs.

6. Binding areas where the constraint satisfaction rules could be defeated are shown below.

In Figure 11, an example is shown where the linearity of a classical constraint satisfaction case is fuzzified/defeated in the considered binding constraint area.

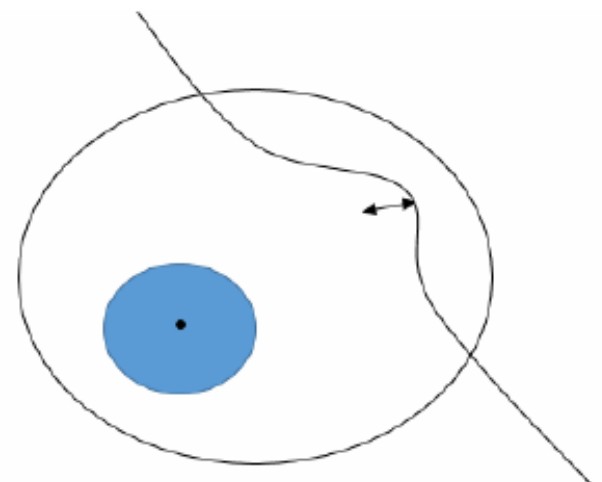

**Figure 11.** Type-6 binding constraint and its influence on classical linear constraints.

Agents, especially in health-oriented systems, can not effectively behave using a set of algorithms. In certain conditions, every solution may be modified or changed. One of the frequent forms of change is defeasible reasoning. In practice, the software agent should defeat its goals aiming at better performance. Every rule should be gradually improved and modified and/or suddenly changed depending on the situation in a data-driven way. The defeasible scheme controls the usage of many unified exclusions and other defeasible knowledge forms.

Let a Horn clause describe Rule (7).

$$B \leftarrow \Lambda_{i \in I} A_i \tag{7}$$

where the form of the rule is suitable for the backward chaining. This rule is changed when an exclusion $E(C, Ak)$ is attached to Equation (7): if $C$ is true, the corresponding $Ak$ is defeated, which means that its truth value is reverted Equation (8) or it disappears from the antecedent Equation (9) because its significance for '$B$ is true' is defeated to zero. Furthermore, the variant Equation (10) is researched where the defeated value is changed by another formula.

$$\frac{B \leftarrow \Lambda_{i=1}^z A_i, C, E(c, A_k), \neg A_k \leftarrow C}{B \leftarrow A_1 \Lambda A_2 \Lambda \cdots A_{k-1} \Lambda \neg A_k \Lambda \cdots A_z} \tag{8}$$

$$\frac{C, B \leftarrow \Lambda_{i=1}^z A_i, E(C, A_k)}{B \leftarrow A_1 \Lambda \cdots A_{k-1} \Lambda A_{k+1} \Lambda \cdots A_z} \tag{9}$$

$$\frac{C, B \leftarrow \Lambda_{i=1}^z A_i, E(C, A_k)}{B \leftarrow A_1 \Lambda \cdots A_{k-1} \Lambda (A_k \forall C) \Lambda A_{k+1} \cdots A_z} \tag{10}$$

One of the frequently used health cases is where both antecedent and consequent are changed in the defeated rule. The other frequent case produces a fact from the defeated rule, and this fact contains a non-implicative relation. The defeasible process explores non-classical rule forms, one of them is '$Ak$ is defeated if $C$ is true in $E(C, Ak)$'. Detailed information concerning this topic is given in book chapter [25].

The defeasible reasoning is applied to test the strength of the investigated process and of its significant features. The quoted binding and pointing constraints significantly improve the defeasible processes. The pointing (indicating) constraints are applied in order to determine both the distance to the goal and the direction of the research. Furthermore, the history of the research process can influence on the pointing direction.

The group of pointing constraints can be considered as a generalization of the classical systems of goal/target or fitness functions. In contrast to the classical cases, pointing constraints are data-driven by nature and revert to being direction-driven with accumulated data. For example, if there is information that there was a pain, the data on its coordinates are probably no longer valid. In this case, the exact conclusion is in doubt until enough proof is accumulated.

The third researched constraint is named 'crossword'. It is depicted by the triple $\{A, C, E\}$ in Figure 10. This type models the process of reasoning on the unknown things based on the accumulated knowledge. In such a way, parts of the searched goal have been found, and by using original evolutionary Puzzle method, a trial to complete the whole solution is attempted. The internal links between the elements of $\{A, C, E\}$ are studied using different combinations of the quoted binding and pointing modeling.

Many algorithms and data-driven approaches were investigated with the aim to find any binding or pointing solution enlarging the set of the known part from $G1$. Generally speaking, every pointing constraint was used aiming to diminish the set of selected/processed data and knowledge: the 'selection focus' should be narrowed. Pointing to a certain part of the binding area improves the reasoning process.

- Type 1: this pointing constraint is based on a priori given special conditions: if $T_1 \star T_2 \cdots \star T_n$ are true, then pointing value of $Y$ is formed, where $\star$ denotes a logical operation conjunction or disjunction.
- Type 2: this pointing constraint is based on conditions: if $T_1 \star T_2 \cdots T_n$ are true, then pointing value of $Y$ is formed, and $T_i$ have been formed in a data-driven manner. $Y$ is gradually changing its direction and value.

On the other hand, in mobile signal processing applications, the pointing constraints show to the center of corresponding binding area. The binding cases frequently concern

best signal processing practice, good medical practice, and many analogical examples. In many cases, little but important signal changes may be traced in this way, where neuro-fuzzy deep learning methods are successfully able to be combined with binding approaches. For example, the pointing constraints have been used for descriptions and maximization of the effectiveness of specific brain–computer interface features and communications to other medical procedures and devices.

Please note that the good practice examples were researched in coordination with possible bad practice situations, where they are described by using a combination of pointing and binding constraints. The considered novel modeling by using constraints does not change, in any direction, the considered biomedical research schemes but improves the effectiveness.

### 9.2. Analysis on Preprocessing and Postprocessing Features

The pointing constraints in this article concern the research on interconnection of electromagnetic and thermal influence, the influence of thermal-located radiation on skin/brain aging, an exploration of possibilities of overreaction to mobile radiation in small groups of people, and the influence of emotional state of researched people to their reaction. Noise in the head, forgetting, loss of concentration, bad mood, and slight disorientation symptoms after the call are not the factors of the binding process but they should be further investigated one by one of in a group with other medical data.

The binding constraints aim to model the facts how the size of the overheated area is connected to the damage effect, why the spots T3, T5, and F7 are the most promising places to estimate the aging affects, and the possible influence of radiation and fields on brain tissue processes, some of them presumably unknown. Other binding modeling options concern the research on mobile radiation in smoke, wet, and dirty air environments, and the binding of high-level SAR signals to the spectral power of EEG/electromagnetic field absorbtion.

Human health problems are a result of long-term dynamic process in nonlinear, and dispersive tissues. Potentially important outcomes may be obtained during long-term research on the accumulated effects on humans after 20, 30, and even 50 years of mobile phone usage. This type of work could not be executed manually, Deep Learning (DL) should be applied instead. The proposed deep modeling approaches should help to trace and process tiny changes and gradually re-evaluate their significance.

Significant ANOVA variables should be reinforced by the proposed types of pointing and binding constraints aiming to improve the constant knowledge elicitation, accumulation and processing possibilities. In this way the principles of open systems had been applied to statistical type of research making it more data-driven. The good practice database contains facts that many people are using hands-free,microphone,Bluetooth and other options where the mobile device is far from the head. This does not mean that the problem has an easy solution: the same sort of radiation still remains just near the human body.

Aiming at long-term search the binding constraints are set to search allergic reactions, oncology-like or blood problems, stomach infections, pain picture, influenza history, and Alzheimer symptoms. The history before and after the active start of the mobile era should be compared. The earlier history of PC usage also is included in the research. As a result, human-like reasoning is inspired: the radiation implies slightly higher temperatures in skin and tissue, and how high should be this difference to be physically noticed by the patient in very cold or cold environment.

This is an experimental attempt to bind the mobile radiation to any noticeable influence on the human body. In the positive case the binding/pointing areas may be enlarged by using other types of constraints. If the patient has any specific problems just after the long calls, he should be analyzed in the lab aiming to bind the problem to mobile radiation as a research hypothesis.

The aging and other human body features depend on many personal factors and history. This is frequently used to oppose to many of the medical research data in the field. As a whole it is rather easy to prove or destroy any hypothesis from the scope in a narrow research concentrated only to one group of facts and features. The proposed deep modeling options helped us to escape from this situation.

The complex research involving DL in perspective will be processed by software agents. In such a way, the types of constraints should be changed depending on the situation. As a result, powerful data science will analyze symptoms in each case by using the accumulated medical data and knowledge. The early eradication of irrelevant medical facts and hypotheses should be preserved by the use of software agents.

The use of the proposed data-science techniques opens up the prospect of greatly reducing manual effort and paving the way for intelligent medical research that focuses on combinations made up of minor and ancillary aspects. Sometimes little details might alter the overall course of the investigation.

## 10. Conclusions

We developed a novel data-science technique to identify the detrimental impacts of electromagnetic radiation from mobile devices on the human body. The proposed method for analyzing heterogeneous data is based on mathematical and statistical methodologies (thermal imaging and electroencephalograms). The proposed solution combines the ANOVA statistical method with deep modeling and rapid preprocessing approaches, such as binding/pointing/crossword constraints. Several tests were conducted utilizing the Pwelch function of MATLAB software, both with and without a mobile device. Each experiment was 30 min long.

The resulting signals were filtered and classified into four frequency ranges: delta (1–4 Hz), theta (4–8 Hz), alpha (8–13 Hz), and beta (13–32 Hz). The accuracy was determined for each frequency range with and without a mobile device based on the collected signals. The findings demonstrate that the emission of electromagnetic radiation from mobile devices has an effect on the signal frequency range accuracy. The presence of irradiation leads to an increase in the amplitude of brain signals in different frequency ranges. Furthermore, the results show that improved accuracy was reached without the use of a mobile device for each frequency band.

The proposed approach has limitations because it increases the computational complexity due to obtaining heterogeneous data. However, this issue can be resolved using a data-mining approach. In the future, different Quality-of-Service parameters (e.g., energy consumption, time complexity, and reliability) will be examined in the future. Furthermore, the proposed approach will also be compared to state-of-the-art approaches: IoT based mobile monitoring framework for hyper-local PM2 [1], and cognitive emotion pre-occupation method [38].

**Author Contributions:** M.G., conceptualization, writing, idea proposal, methodology, and results; V.J. and I.G., data curation, software development, and preparation; A.R., Writing, results, software development, preparation, submission, review and editing; M.A. and B.A., review, manuscript preparation, and visualization; D.B. review and editing. All authors have read and agreed to this version of the manuscript.

**Funding:** This work is supported by the Bulgarian National Science Fund, project title "Synthesis of a dynamic model for assessing the psychological and physical impacts of excessive use of smart technologies", KP-06-N 32/4/07.12.2019, led by Magdalena Garvanova.

**Institutional Review Board Statement:** Not applicable.

**Informed Consent Statement:** Not applicable.

**Data Availability Statement:** The data that supports the findings of this research is publicly available as indicated in the reference.

**Acknowledgments:** This work was supported by the Bulgarian National Science Fund, project title "Synthesis of a dynamic model for assessing the psychological and physical impacts of excessive use of smart technologies", KP-06-N 32/4/07.12.2019, led by Magdalena Garvanova.

**Conflicts of Interest:** The authors declare no conflict of interest.

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
