# Peer review of "A Data-Science Approach for Creation of a Comprehensive Model to Assess the Impact of Mobile Technologies on Humans"

_applsci, doi:10.3390/app13063600_

Round 1

Reviewer 1 Report

The authors used data science techniques to access the impact of mobile technologies on humans. 

The paper is very interesting and this address a real issue with all humans. Although the problem addressed in this article is genuine, there are many types of research that guide us about the disadvantages of using mobile phones for long period.  

The abstract size is very large. I think the maximum number of words in the abstract must be less than 300.

Section 2 is about  Statistical Data Processing from Empirical Research. What is the rational to include this section in this article. Empirical research has different and many protocols that were not discussed in this research.  Have the authors conducted empirical research in this article?

The article has no literature Review Section in the paper. Please add a Literature review section in the paper to prove that before and during the writing of the paper, the authors have studied the concerned and the latest research.

Please add a Methodology section in the paper and include relevant material in this section.

The results are required to be placed under the heading of the Results and Discussion Section. Please discuss each result properly.

In Conclusion, the authors did not add limitations to the study and future direction. Please add these things in the last section of the paper.

I think the authors used some referencing tools for referencing. Various references are incomplete. Please revisit all the references to add missing details.

Reviewer 2 Report

This research evaluates the impact of mobile technologies on humans using a data science approach. Considering the fruitful discussion, the present literature review could be extended further, and this manuscript could become concise to be more engaging for the reader. Furthermore, the manuscript requires proofreading. The further comments are as follows:

1. The study needs to include the relevant literature from 2020 to 2023 and mention how the present study improves the previous work and contributes to the research gaps.

2. The manuscript should be more concise. For example, there are some avoidable repetitions in section 5. Moreover, section 8.1. Deep knowledge Modeling… should be summarized while addressing the objective of the study. Lines 120 to 140 related to the following sections of the study could also be summarized.

3. The conclusion could be further expanded. In line 699, please provide more details about the effect of the electromagnetic radiation on the signal accuracy. In lines 701 and 702, please explain further the different Quality-of-Service parameters that will be examined. In line 703, please mention some of the state-of-the-art approaches.

4. In 7.1, the Figure mentioned in the text should be 9, not 8. 

5. Please improve the title of Fig 9. Perhaps start with "The average spectrum exposure with and without GSM.." and then explain sections a to d. 

6. Regarding reference 37, please avoid using references from non-peer-reviewed depositories like arXiv. 

7. Please provide the link for the source code to help other scholars reproduce the experiment. 

8. Please double-check all equations and ensure that their parameters are adequately explained.

9. Please add a brief sentence to concisely explain the sub-sections of section 8.

10. The current submission requires proofreading, as some incomplete, and unclear sentences hinder demonstrating the importance of the work. I highlighted some concerns about the English language in the file attached.

11. For other corrections, please refer to the attached file.

12. Please highlight the corrections for my comments in the revised file to speed up the review process.

All the best.

Round 2

Reviewer 1 Report

I have checked the replies of the authors and revised the manuscript. The authors have not given a satisfactory reply against comments no. 2.

Although i am approving this article, the authors should again check the use of statistically related material.  

Reviewer 2 Report

The comments have been mainly addressed. The paper could be further reorganized and the language could be improved by proofreaders or application like Grammarly to help the paper sounds scientific. I would recommend reading this interesting book to reorganize the work:

Barbara Gastel, Robert A. Day - How to Write and Publish a Scientific Paper-Greenwood (2022)

Others: 

The paragraph in 2. Related Work is lengthy. Perhaps break it down into different paragraphs to engage the reader. 

Figure 8's title is repetitive. Please summarize it. 
